

# Methane flux estimates from continuous atmospheric measurements and surface-water observations in the northern Labrador Sea and Baffin Bay

Judith Vogt[1,2], David Risk[1], Kumiko Azetsu-Scott[3], Evan N. Edinger[4], Owen A. Sherwood[5]

[1]Department of Earth Sciences, St. Francis Xavier University, Antigonish, B2G2W5, Canada
[2]Environmental Science Program, Memorial University of Newfoundland, St. John's, A1B3X7, Canada
[3]DFisheries and Oceans, Bedford Institute of Oceanography, Dartmouth, B2Y4A2, Canada
[4]Department of Geography, Memorial University of Newfoundland, St. John's, A1B3X9, Canada
[5]Department of Earth and Environmental Sciences, Dalhousie University, Halifax, B3H4R2, Canada

*Correspondence to*: Judith Vogt (jvogt@mun.ca)

**Abstract.** Vast amounts of methane ($CH_4$) stored in permafrost and submarine sediments are susceptible to release in a warming Arctic, further exacerbating climate change in a positive feedback. It is therefore critical to monitor $CH_4$ over pan-regional scales to detect early signs of $CH_4$ release. However, our ability to monitor $CH_4$ is hampered in remote northern regions by sampling and logistical constraints and few good baseline data exist in many areas. To create a baseline study of current background levels of $CH_4$ in North Atlantic waters, we collected continuous real-time atmospheric $CH_4$ data, along with ambient air temperature and wind parameters over 22 days in summer 2021 on a roughly 5100 km voyage in the northern Labrador Sea and Baffin Bay up to 71°N. In addition, we measured $CH_4$ concentrations in the water column using discrete water samples at selected stations. Measured atmospheric mixing ratios of $CH_4$ ranged from 1944.7 ppb to 2012.0 ppb, with a mean of 1966.0±7.4 ppb and a baseline of 1954.2−1980.6 ppb. Dissolved $CH_4$ concentrations in the near-surface water peaked at 56.58±0.05 nM within 1 km down-current of a known cold seep at Scott Inlet but were consistently super-saturated throughout the water column in Southwind Fjord, which is an area recently affected by submarine landslides. Local sea-air $CH_4$ fluxes ranged from 0.1−14.1 µmol m$^{-2}$ d$^{-1}$ indicating that the ocean acted as a $CH_4$ source to the atmosphere. Atmospheric $CH_4$ levels were also driven by meteorological, spatial, and temporal variations. Highest atmospheric $CH_4$ mixing ratios were detected in the Cumberland Sound in Nunavut, suggesting onshore sources from nearby waterbodies and wetlands, whereas ocean-based contributions at this location could not be ruled out. Coupled real-time measurements of marine and atmospheric $CH_4$ data have the potential to provide ongoing monitoring in a region susceptible to $CH_4$ releases, as well as critical validation data for global-scale measurements and modelling.

## 1 Introduction

Global atmospheric methane ($CH_4$) levels have substantially increased in recent years, with the largest recorded yearly increase from 2020 to 2021 (Dlugokencky, 2016; Nisbet et al., 2019). Due to the high radiative activity of the greenhouse gas $CH_4$,





close observations of atmospheric levels are needed to determine trends and impacts on the future climate. While Arctic regions are subject to rapid warming (Meredith et al., 2019), measurements of atmospheric $CH_4$ levels in these regions are scarce, especially over the ocean. The Arctic Ocean contains large amounts of $CH_4$ in sediments along the continental margins. With ongoing climate change, permafrost thaw, destabilization of $CH_4$ hydrates and reduction of sea ice cover may make the Arctic

Ocean susceptible to substantial $CH_4$ release further exacerbating global warming (James et al., 2016). Seafloor gas seeps releasing $CH_4$-rich bubbles into the water column are often found along continental margins. However, the contribution of seafloor gas seeps to atmospheric $CH_4$ entails large uncertainties (Saunois et al., 2016), mostly due to significant temporal and spatial differences of emissions (Boles et al., 2001; Leifer and Boles, 2005; Shakhova et al., 2014; Cramm et al., 2021; Dølven et al., 2022). Water depth and the abundance of methanotrophic bacteria influence the oxidation of $CH_4$, and the speed and

strength of currents affects the dissolution of the gas (McGinnis et al., 2006; Reeburgh, 2007; Leonte et al., 2017; Silyakova et al., 2020). Among others, these factors determine how much of the gas diffuses to the atmosphere.

While the East Siberian Arctic Shelf overall releases up to 4.5 Tg $CH_4$ yr$^{-1}$ of $CH_4$ of mostly thermogenic, but also biogenic origin (Berchet et al., 2020) with large temporal and spatial variability (Shakhova et al., 2010, 2014; Thornton et al., 2016, 2020), prevailing thought suggests that the North American Arctic Ocean contributes relatively little $CH_4$ to the atmosphere

(Manning et al., 2022). Increasing atmospheric concentrations of $CH_4$ have however been reported over the European Arctic Ocean and mostly attributed to land-based sources, but also marine point-sources from active underwater seeps (Platt et al., 2018). While a few studies focused on dissolved $CH_4$ levels in north-eastern Canadian Arctic waters (Punshon et al., 2014, 2019) where seep locations were suggested (Jauer and Budkewitsch, 2010; Punshon et al., 2019) or confirmed (Cramm et al., 2021), continuous measurements of atmospheric $CH_4$ levels in this region are lacking and more measurements in this area are

needed. To investigate how the identified seep areas affected atmospheric $CH_4$ levels, we conducted mobile $CH_4$ monitoring onboard the icebreaker CCGS *Amundsen*. We collected measurements of $CH_4$ dissolved in the water column at select locations between the northern Labrador Sea to Baffin Bay adding to a small but growing body of data on water column $CH_4$ concentrations in the Arctic and sub-Arctic seas. We also tracked atmospheric $CH_4$ levels continuously along a north-south transect to establish a baseline study for above-ocean $CH_4$ mixing ratios in the area that can be used as a benchmark for further

monitoring of $CH_4$ levels in Arctic regions.

## 2 Methods

### 2.1 Study area

Data for this study was collected during an expedition of the Canadian research icebreaker CCGS *Amundsen* starting on July 15, 2021, in St. John's, Newfoundland, Canada, and ending on August 12, 2021, in Iqaluit, Nunavut, Canada. The expedition

transited the western Labrador Sea, Davis Strait, and Baffin Bay along the north-eastern Canadian continental shelf (Fig. 1). Along the shelf margins, seafloor gas seepage was previously localized at Scott Inlet, Baffin Bay (71.37812 N, −70.07452 W) (Loncarevic and Falconer, 1977; Levy and MacIean, 1981; Cramm et al., 2021), while further locations were suggested in the



Saglek Basin in northern Labrador (60.351 N, −61.864 W) (Jauer and Budkewitsch, 2010; Punshon et al., 2019) and off the
coast of Cape Dyer, Baffin Island (67.449 N, −61.919 W) (Punshon et al., 2019). The studied region lies within the seasonal
sea ice zone and the ocean was partially covered with sea ice in the northernmost regions. Hydrography in the studied area is
dominated by the Baffin Island Current (BIC). The BIC is the integrated Arctic outflow through the Canadian Arctic
Archipelago, flowing southward along the Baffin Island coast and slope. The BIC becomes a component of the Labrador
Current, being modified by the Hudson Strait overflow, and continues flowing southward, mainly confined to the shelf and
upper slope (Azetsu-Scott et al., 2012). The West Greenland Current bifurcates at Davis Strait, with part of the flow entering
Baffin Bay on the eastern side of Davis Strait and contributing to the cyclonic circulation in the Bay, and partly continuing
westward as the Labrador Sea cyclonic circulation (Melling et al., 2001; Tang et al., 2004; Wu et al., 2013). The eastern coast
of Baffin Island is characterized by the Baffin Mountains, with elevations up to 2147 m. With its location north of the tree line,
the land is dominantly barren and sparsely vegetated, or covered with smaller waterbodies and wetland areas.





**Fig. 1: The ship's trajectory and atmospheric CH₄ levels as averages over consecutive 10 km sections. The black arrows point to the locations where water measurements were taken. The three black hexagons indicate confirmed or suspected locations of gas seepage (Punshon et al., 2014, 2019; Cramm et al., 2021). White arrows represent the West Greenland Current (WGC) and Baffin Island Current (BIC). Water depth was retrieved from the NOAA server (Amante and Eakins, 2009). Areas labelled a, b and c indicate the extents for each panel in Fig. 7.**

**2.2 Atmospheric measurements**

Instruments were mounted on the Meteorological Tower at the bow of the ship: A 2D heated anemometer (Model 86004, RM Young, USA) at a height of 8.1 m above deck, a temperature probe (Model 107B, Campbell Scientific, USA) 7.6 m above the deck, a 1 Hz GPS puck (GPS 18x LVC, Garmin, USA), and an air inlet for gas sampling at 7.3 m (Appendix A, Fig. A1).



Roughly 30 m long Synflex tubing connected the air inlet with the greenhouse gas analyzer (Ultraportable Greenhouse Gas

Analyzer, Los Gatos Research, USA), making real-time monitoring of atmospheric carbon dioxide ($CO_2$), methane ($CH_4$) and water vapor ($H_2O$) mixing ratios possible. The analyzer is equipped with a built-in pump drawing the air from the inlet on the tower to the analyzer stored securely inside a laboratory on deck. The greenhouse gas analyzer was calibrated in July 2021 before deployment on the ship with certified calibration gas (385.18±0.01 ppm $CO_2$, 1810.6±0.1 ppb $CH_4$, 4.08±1.58 ppm $H_2O$), and benchmarked daily (except for the first two days due to logistical issues) with a certified standard gas mixture of

450 ppm $CO_2$ balanced with air containing 5000 ppb $CH_4$, which was well within the analyzer's measurement range (200−20,000 ppm for $CO_2$ and 100−100,000 ppb for $CH_4$). Once the setup was mounted and leak proof, we recorded atmospheric measurements over a distance of 5100 km between July 20, 2021, and August 10, 2021, on a datalogger (CR1000, Campbell Scientific, USA) at a frequency of 1 Hz.

We pre-processed the obtained data by excluding inconclusive values of position, wind direction and speed. Resulting gaps

and missing values of mixing ratios were linearly interpolated, corresponding to 19% of all sampled 1 Hz data for gas mixing ratios. By repeatedly breathing on the air inlet, we determined an average delay time of 90 seconds for the air samples to reach the analyzer and accounted for this delay time during pre-processing. Wind parameters were corrected for lateral ship motion when the ship was not in transit or not headed forward, using speed, track and heading from the ship's own navigation system (Amundsen Science Data Collection, 2021c). The gas analyzer did not significantly drift over time (in comparison to the

manufacturer's precision specification 2 ppb for 1 σ), and we assessed instrument noise and drift in combination by integrating the data from benchmarking while on the ship and determined a standard error of 2.1 ppb for $CH_4$ and 0.13 ppm for $CO_2$ that can be considered the uncertainty of our measurements. In addition, we determined there was no significant contamination of air samples by considering $CO_2$ mixing ratios when the air inlet was downwind of the ship's (comparatively elevated) exhaust. To determine $CH_4$ baseline levels for the studied region, we applied a Savitzky-Golay filter (Savitzky and Golay, 1964) of

second polynomial order on the mixing ratios.

Maxima in atmospheric $CH_4$ measurements were further investigated using the online Real-time Environmental Applications and Display System (READY) for the Hybrid Single-Particle Lagrangian Integrated Trajectory (HYSPLIT) model (Stein et al., 2015; Rolph et al., 2017). Ensemble back-trajectories of air masses from the time and location where $CH_4$ maxima were measured (referred to as source) to the point of possible origin within the previous 12 hours were modelled. Two gridded

meteorological data archives were used: the Global Data Assimilation System (GDAS) model (1° horizontal resolution) and the Global Forecast System (GFS) model (0.25° horizontal resolution). For the ensemble, the datapoints of the meteorological input model were offset by a fixed grid factor resulting in an output of 27 possible trajectories (Rolph et al., 2017).

Atmospheric pressure and dew point temperature measurements were recorded every two minutes with a digital barometer (PTB-210, Vaisala, Finland) and a humidity-temperature sensor (MP101A-T7, Rotronic, USA) located on the bridge of the

ship (Amundsen Science Data Collection, 2021b). For statistical analyses, we log-transformed the non-normally distributed $CH_4$ mixing ratios, and also fitted a simple Generalized Additive Model (GAM; used previously in air quality studies, e.g. Pearce et al., 2011; Hou and Xu, 2022) to hourly averaged $CH_4$ data in order to identify trends of inter-dependencies. The



GAM was well suited due to its ability to describe non-linear effects of non-normally distributed data using non-parametric smoothing functions. The respective analysis was performed in R (package: "mgcv", function: "gam"; Wood, 2011).

## 2.3 Water column measurements


Seawater was collected at 13 stations for measurements of dissolved $CH_4$: north-eastern Labrador ("Kelp"), two locations at Saglek Bank, Hatton Sill, Davis Strait, Southwind Fjord, Disko Fan, five locations at Scott Inlet, and Clark Fiord (Fig. 1 and 2). While exclusively surface samples were taken at Clark Fjord and at four co-located stations close to the Scott Inlet seep (SI1, SE-1K, NE-1K, NE-5K), we gathered water column profiles at the remaining eight locations. Collection and
measurement protocols followed that of Punshon et al. (2014, 2019). Briefly, seawater from discrete depths was collected into 12 L Niskin bottles mounted on a Conductivity-Temperature-Depth (CTD) Rosette. On recovery, the waters were transferred to 60 ml glass serum bottles (after triple rinsing with the sample water) to overfilling, immediately fixed with mercuric chloride, capped with metal crimp seals and rubber septa, and stored at 4°C. Samples were analyzed for $CH_4$ at the Bedford Institute of Oceanography (Department of Fisheries and Oceans, Canada) using a single-phase batch headspace equilibration method with
gas chromatography (similar to Neill et al., 1997). Uncertainty in dissolved $CH_4$ was ±0.08% (Punshon et al., 2014, 2019). Data from previous studies conducted in 2011, 2012 and 2016 (Punshon et al., 2014, 2019) were included here to examine regional patterns and temporal variations of dissolved $CH_4$ concentrations in the Baffin Bay. Potential temperature ($\theta$) and density of seawater at atmospheric pressure ($\sigma_\theta$) were calculated based on water temperature, pressure and salinity measured on the ship (SBE 9plus CTD, Seabird Scientific, Canada) (Amundsen Science Data Collection, 2021d) using the package
'seawater' in Python (calculations based on Bryden, 1973; Fofonoff and Millard, 1983; Millero and Poisson, 1981). Water masses were assigned according to operational definitions considering specified ranges of calculated potential temperature and density of seawater (Table 1 in Sherwood et al., 2021; Stramma et al., 2004; Fratantoni and Pickart, 2007; Azetsu-Scott et al., 2012). These water masses comprise Halocline Water ($\sigma_\theta \le 27.30$ kg/m³, $\theta \le 0$°C), Baffin Bay Water ($27.50 < \sigma_\theta \le 27.80$ kg/m³, $\theta \le 2$°C), Labrador Shelf Water ($\sigma_\theta \le 27.40$ kg/m³, $\theta \le 2$°C), Irminger Water ($27.30 < \sigma_\theta \le 27.68$ kg/m³, $\theta > 2$°C), Labrador
Sea Water ($27.68 < \sigma_\theta \le 27.80$ kg/m³, $\theta > 2$°C), and to a lesser extent North East Atlantic Sea Water ($27.80 < \sigma_\theta \le 27.88$ kg/m³) and Denmark Strait Overflow Water ($\sigma_\theta \le 27.88$ kg/m³). It should be noted that surface waters (~2 m) did not necessarily match operational definitions of water masses as outlined in Sherwood et al. (2021) and were interpreted separately.



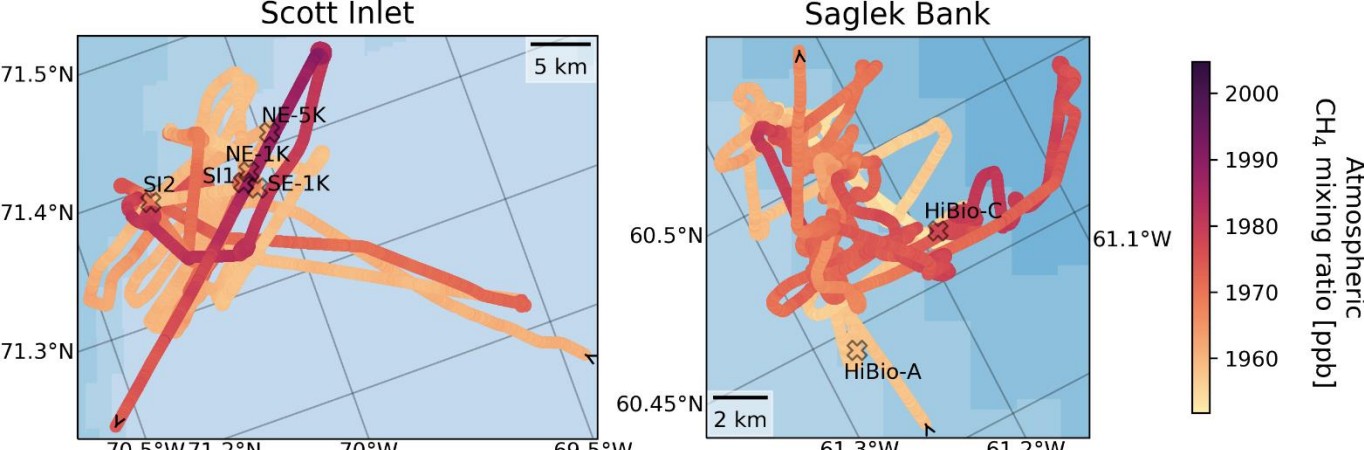

**Fig. 2: Close-up of Scott Inlet and Saglek Bank, where multiple water measurements were taken. The locations of CTD-Rosette**
**sampling are indicated together with respective names of stations. The arrows indicate the direction where the ship was heading.**
**Station SI1 was located at the seep at Scott Inlet (left panel).**

## 2.4 Sea-air methane flux

The sea-air $CH_4$ flux (F) was determined with the bulk flux equation (Wanninkhof, 2014),

$$F = k\,(C_w - C_a),$$

combining measured dissolved $CH_4$ concentrations ($C_w$) and air-equilibrated seawater $CH_4$ concentrations ($C_a$) (Equation 7,
Wiesenburg and Guinasso, 1979) calculated with our atmospheric $CH_4$ measurements averaged between three minutes before
and after the time of sampling, as well as water temperature and salinity measurements from the CTD. The gas transfer velocity
(k) (Wanninkhof, 2014) was determined with

$$k = 0.251\,\overline{U}^2\,(Sc/660)^{-0.5},$$

making use of the Schmidt number (Sc) (Table 1, Wanninkhof, 2014) and wind speeds averaged between three minutes before
and after the time of sampling ($\overline{U}$).

## 3 Results and discussion

Seawater samples showed wide ranges of dissolved $CH_4$ concentrations at the different sample locations and water depths
from undersaturated (53%, 0.2 nM) to highly oversaturated (6858%, 272.4 nM, Fig. 3). The highest water column
concentrations were measured about 8 km north-west of the known cold seep at Scott Inlet (station SI2, Fig. 2) at 200 m water
depth, decreasing to 1213% (41.8 nM) at the surface. These high concentrations were not surprising given documented
ebullition observed previously in the area (Cramm et al., 2021). Measurements from the year 2012 revealed $CH_4$ maxima of
65.8 nM at 200 m depth decreasing to 3.7 nM at the surface roughly 40 km north-west from the seep location (Punshon et al.,
2019). Large temporal fluctuations of dissolved $CH_4$ levels between 9 and 609 nM within 24 hours were found close to the
seafloor (~250 m) at the seep in 2018 (Cramm et al., 2021). Similarly, other studies also manifested the temporal variability



of seafloor seep degassing (Boles et al., 2001; Leifer and Boles, 2005; Shakhova et al., 2014; Cramm et al., 2021; Dølven et al., 2022). However, concentrations at the water surface of the seep were in the single digits in the past (Cramm et al., 2021). Where high bottom concentrations within 5 km of the seep were measured in Cramm et al. (2021), we found elevated concentrations between 42.7 nM (station NE−5K, roughly 5 km north-east of the seep) and 56.9 nM (station SE−1K, about 1
km south-east of the seep) at the water surface (Fig. 2 and 3). Considering the findings from Punshon et al. (2019), Cramm et al. (2021) and the present study, depths of ~200 m around the Scott Inlet seep location seemed most prominent for $CH_4$ maxima. Furthermore, elevated $CH_4$ concentrations at this location over several years shows the persistence of the seep activity. Surface concentrations an order of magnitude higher in 2021 may indicate reduced oxidation of $CH_4$ within the water column relative to other years. However, this station should not be considered as representative of the Baffin Bay as a whole, but rather specific
to the seep location.

The second highest concentrations of dissolved $CH_4$ were measured at Southwind Fjord with a maximum of 2418% oversaturation (93.8 nM) at 30 m depth, 1578% (55.7 nM) at the surface, and 1210% (48.0 nM) at the bottom (75 m). Occurrences of highly supersaturated waters in Artic and sub-Arctic fjords have been documented previously: up to 33.5 nM
and 974% super-saturation in the Isfjorden, Svalbard, Norway (Damm et al., 2021), up to 72.3 nM and ~2000% super-saturation in the Storfjorden, Svalbard, Norway (Mau et al., 2013) and up to 459.2 nM at the head of the Canadian sub-Arctic Saguenay fjord (Li et al., 2021). Possible sources of high dissolved $CH_4$ concentrations at Southwind Fjord in this study could be terrestrial runoff, although Manning et al. (2022) found that rivers did not discharge significant amounts of $CH_4$ to the North American Arctic Ocean in the summers of 2017−2019. Advection of $CH_4$-rich water from other sources within the Baffin Bay
could play an important role given the evidence of oil slicks off Cape Dyer for example (Budkewitsch et al., 2013). Other potential sources could be unknown seeps within the fjord or the recent disturbance from iceberg groundings and subsequent landslides (Normandeau et al., 2021), which could have led to $CH_4$ release into the water column from a fresh supply of organic matter, or gas hydrates or $CH_4$-bearing pore water in the seafloor sediment disturbed by the turbulence (Paull et al., 2002). Overall, we recommend follow-up sampling to assess the persistence of the $CH_4$ super-saturation and its source at Southwind
Fjord.





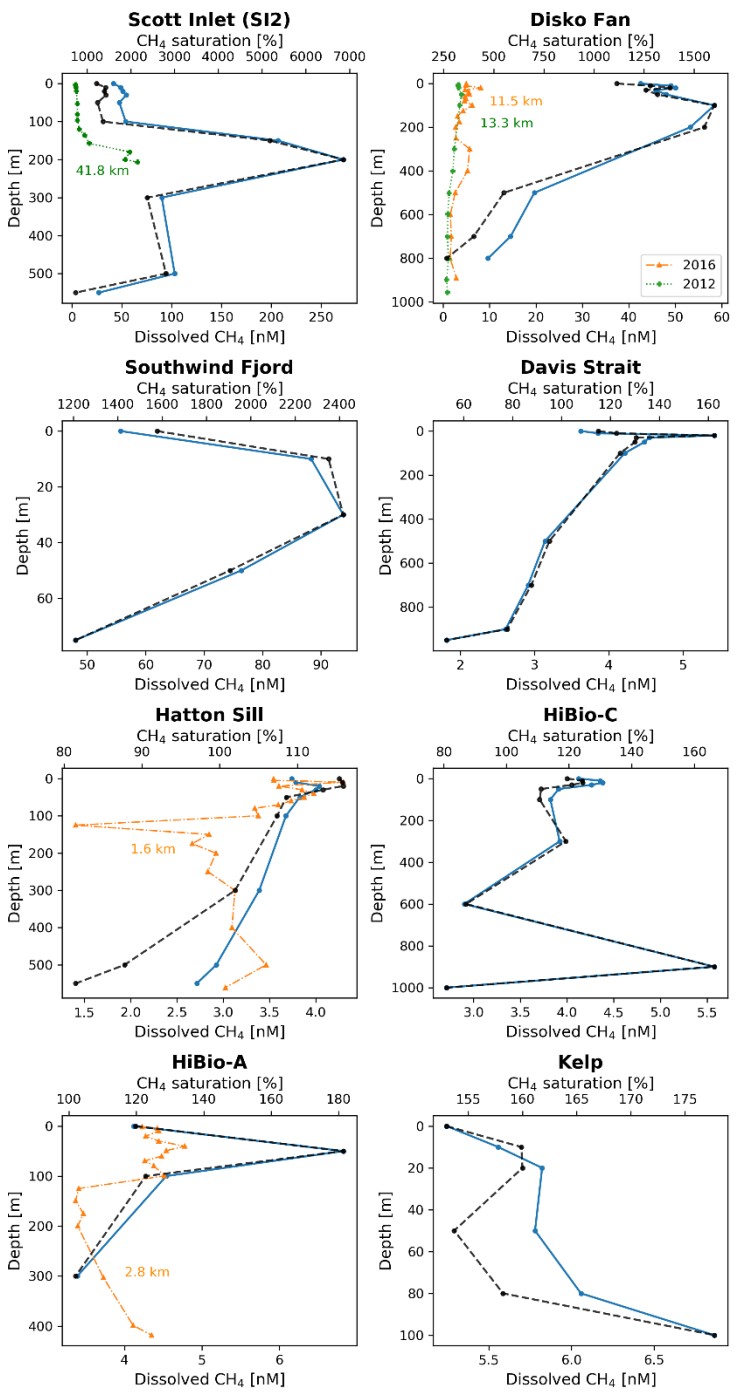

**Fig. 3: Depth profiles of dissolved CH₄ concentrations (blue) and saturations (black, dashed line) throughout the water column. Station names are given and can be located on Fig. 1 and 2. Profiles from Punshon et al. (2014, 2019) conducted in 2012 and 2016 were included for stations close to the ones from 2021 and are shown in green (2012) and orange (2016). Distances between respective nearby stations are given in kilometres.**





The Disko Fan station had concentrations of dissolved $CH_4$ that were the third highest in this study. The $CH_4$ maximum with a super-saturation of 1600% (58.4 nM) at 100 m depth, decreased with depth to 265% (9.7 nM) at 800 m, and remained high (>1000%) towards the surface. These results were much higher than measurements from nearby stations in 2012 and 2016 (Punshon et al., 2019) as shown in Fig. 3, and an order of magnitude higher than measurements on cross-basin transects

(Punshon et al., 2014). While the cause of this increment remains unknown, prevailing currents (West Greenland Current) and the shallow depth of the $CH_4$ maximum (100 m) suggest a $CH_4$ source around the south-western Greenland shelf, where possible $CH_4$ seepage was suggested (Gregersen and Bidstrup, 2008; Gautier et al., 2011; Nielsen et al., 2014). Moreover, onshore lakes in south-west Greenland showed highest dissolved $CH_4$ concentrations (average of 2530 nM) among all reported lakes at northern latitudes (Northington and Saros, 2016), and glacial runoff from the Greenland ice sheet caused $CH_4$ discharge

of an average 271 nM (Lamarche-Gagnon et al., 2019). Alternatively, increased $CH_4$ levels could originate from an extension of $CH_4$-rich water spreading from the western side of Baffin Bay. These findings also warrant the need of continued monitoring to see if high $CH_4$ levels at this location are persistent.

All other stations from Davis Strait and further southward along the northern Labrador shelf showed significantly lower

dissolved $CH_4$ concentrations than any of the Baffin Bay stations. Respective subsurface maxima showed over-saturation between 116−181% (4.0−6.9 nM) at varying depths. $CH_4$ concentrations at these stations except for "Kelp" tended to decrease with depth, suggesting advection of $CH_4$ within shallow water masses from elsewhere. Compared to measurements at nearby locations in 2016, dissolved $CH_4$ concentrations in 2021 at the stations Hatton Sill and HiBio-A were very similar ranging between 2.7−6.8 nM (Fig. 3). Average water column $CH_4$ concentrations of stations south of 65°N in 2021 (mean: 4.2 nM,

range: 1.8−6.9 nM) were close to those measured in the Davis Strait in previous years (mean: 3.9 nM, range: 1.1−10.5 nM; Punshon et al., 2014, 2019).

The distribution of $CH_4$ with respect to water masses accounting for data from Punshon et al. (2014, 2019) and this study are visualized in a temperature-salinity diagram (Fig. 4). Samples span the known upper and intermediate depth of water masses

of the region, mainly Halocline Water (HW), followed by Irminger Water (IM), Labrador Shelf Water (LShW) and Baffin Bay Water (BBW). Highest concentrations were found in Arctic HW (mean: 12.3 nM, range: 2.4−272.4 nM), which was partly forced by the presence of HW overlying most of the water column at the Scott Inlet seep. This seep, and possibly others, could enrich the HW with $CH_4$ as HW travels southward in form of the Baffin Island Current. The overall shallowest water mass, the LShW, held second highest $CH_4$ concentrations (mean: 6.9 nM, range: 1.1−88.3 nM) possibly due to the influence of the

Baffin Island Current transporting $CH_4$-rich water southward or of the West Greenland Current carrying elevated $CH_4$ levels westward, which may have provoked higher-than-expected $CH_4$ levels in LShW at the Disko Fan station. Warmer IW masses tended to have lower concentrations (mean: 3.5 nM, range: 1.3−53.2 nM), potentially due to increased oxygen availability in the Irminger Sea as found in 2015 (Fröb et al., 2016) which could have led to $CH_4$ oxidation and reduced $CH_4$ levels, for example at the 2021 stations HiBio-A, HiBio-C, Hatton Sill and Davis Strait. Similarly, the colder and deeper BBW mass



showed lower CH$_4$ concentrations (mean: 3.2 nM, range: 0.2−103.2 nM), whereas measurements in proximity to the Scott Inlet seep and at the Disko Fan station in 2021 contributed to the high end (>17 nM) of the concentration range. Therefore, both CH$_4$ production and consumption co-occurred in the BBW (Fenwick et al., 2017).

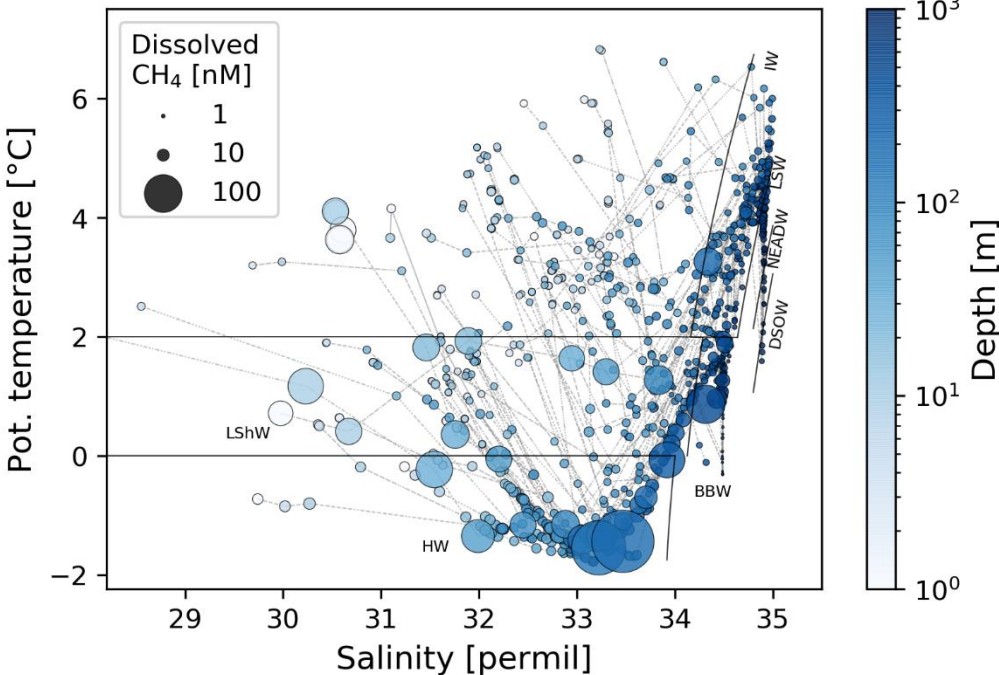

**Fig. 4: Temperature-salinity diagram of all measurements from 2021 and from the studies by Punshon et al. (2014, 2019) for the Baffin Bay and Davis Strait area. Dissolved CH$_4$ concentrations are shown with different marker sizes, colors indicate the water depth. Lines distinguish between water masses: Halocline Water (HW), Labrador Shelf Water (LShW), Irminger Water (IW), Labrador Sea Water (LSW), Northeast Atlantic Deep Water (NEADW) and Denmark Strait Overflow Water (DSOW). Gray lines connect measurements from the same CTD-Rosette cast. For better visualization, salinities below 28‰ which were measured at the two fjords in 2021 are not shown.**

In 2021, CH$_4$ concentrations decreased from subsurface maxima towards the surface at all stations (Fig. 3), which was most likely caused by oxidation within the water column. Nevertheless, surface water concentrations were above saturation at all stations (including further locations around the Scott Inlet seep and at Clark Fiord where only surface samples were taken, Fig. 5). While dissolved CH$_4$ concentrations at latitudes below 65°N ranged from 3.6−5.3 nM, concentrations were one order of magnitude higher in the sampled areas north of 65°N (41.8−56.9 nM). We suggest that differences in surface ocean current patterns with stronger influence of the West Greenland Current joining the Labrador Current (Tang et al., 2004; Curry et al., 2011) maintained lower concentrations of dissolved CH$_4$ at the water surface than above 65°N. Moreover, partial sea ice cover may also have reduced the diffusion of CH$_4$ from surface waters into the atmosphere in higher latitudes (Damm et al., 2015). We observed brief periods of close pack ice, but mostly brash ice between north of 65°N and south of Scott Inlet. High dissolved CH$_4$ concentrations at the surface of the (ice-free) Clark Fiord indicate that CH$_4$-rich water from waterbodies onshore





could have been discharged into the narrow inlet (Manning et al., 2020; Li et al., 2021), or that other processes were responsible

for the $CH_4$ accumulation.

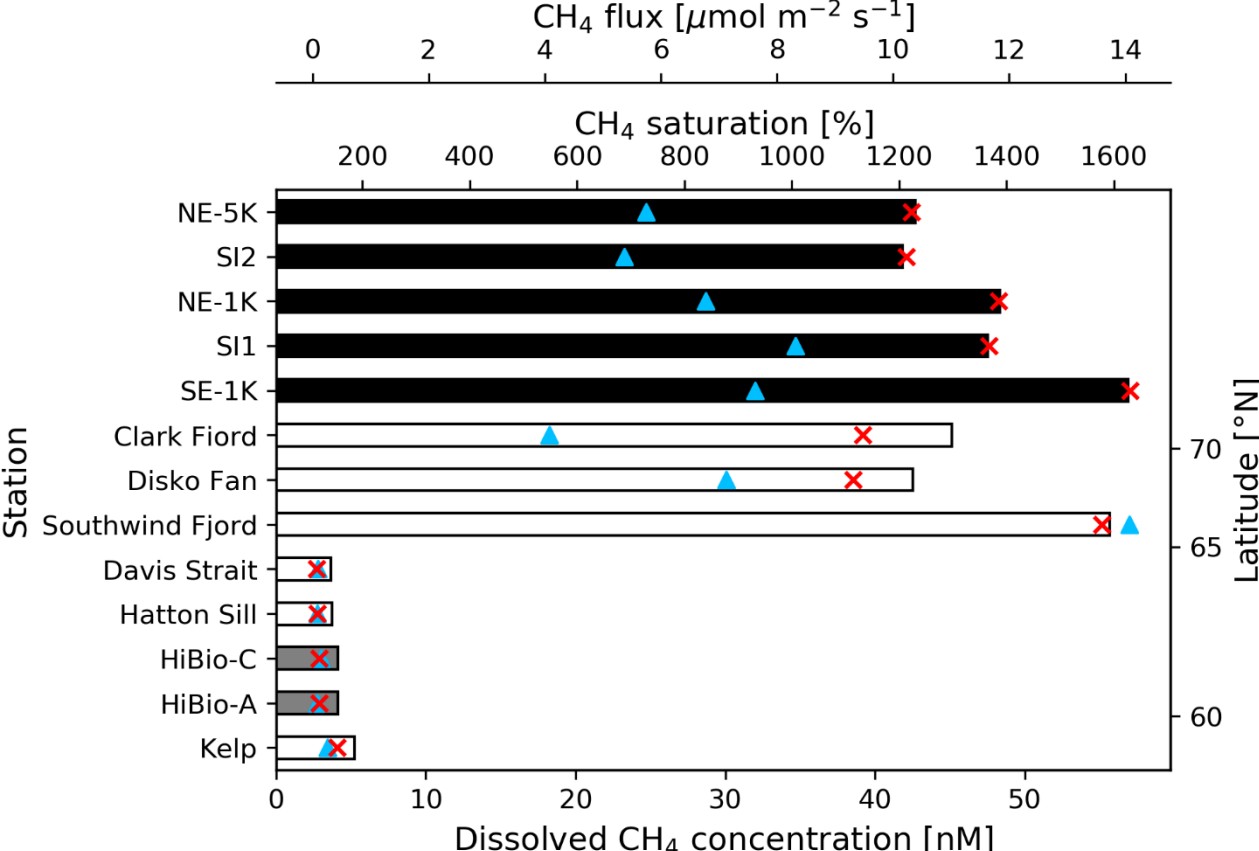

**Fig. 5: Dissolved CH₄ concentrations at the water surface (bars) for all stations where CTD-Rosette samples were collected. Gray bars represent two sample locations in the Saglek Bank area, and black bars reflect samples in the Scott Inlet area, both close to**
**seafloor seep locations (station names correspond to those in Fig. 2). Concentrations north of 65°N were substantially higher than south of 65°N. CH₄ saturations (red crosses) and estimated sea-air fluxes (blue triangles) are shown as well. Latitudes are not to scale.**

In this study, we recorded a net flux of $CH_4$ from the ocean to the atmosphere, which amounted to a mean of 4.6±4.3 µmol m$^{-2}$ d$^{-1}$ based on measurements in 2021, with a mean of 0.1±0.1 µmol m$^{-2}$ d$^{-1}$ for measurement locations south of 65°N, and

7.4±3.0 µmol m$^{-2}$ d$^{-1}$ north of 65°N featuring large uncertainties. Overall, sea-air fluxes in this study peaked at 14.1 µmol m$^{-2}$ d$^{-1}$ in the Southwind Fjord, exceeding the flux rate of 5.4−8.3 µmol m$^{-2}$ d$^{-1}$ generated from the Scott Inlet seep (Fig. 5). As a result, fluxes in the northern Labrador Sea were negligible in summer 2021, whereas mean emission rates in the Baffin Bay beyond 65°N were of similar magnitude as mean estimates of 8.7 µmol m$^{-2}$ d$^{-1}$ for the Chukchi Sea (Thornton et al., 2020), and exceeded averages found in other studies of 1.6 µmol m$^{-2}$ d$^{-1}$ for the Davis Strait (Punshon et al., 2014), 1.3 µmol m$^{-2}$ d$^{-1}$

for the Bering Sea to Baffin Bay (Fenwick et al., 2017), and 0.4 µmol m$^{-2}$ d$^{-1}$ for the Baffin Bay and Davis Strait (Manning et al., 2022). Considering all measurements from 2021 and an area of 1,123,000 km² for the Baffin Bay and Davis Strait (Manning





et al., 2022), we calculated a basin-wide mean net $CH_4$ flux of 0.030±0.029 Tg/yr (median: 0.035 Tg/yr, 25th percentile: 0.001 Tg/yr, 75th percentile: 0.047 Tg/yr). If samples with high concentrations were excluded, the net flux decreases to 0.021±0.036 Tg/yr (median: 0.001 Tg/yr, 25th percentile: 0.001 Tg/yr, 75th percentile: 0.024 Tg/yr), in which case the ocean may act as a

small $CH_4$ source or sink to the atmosphere. Both flux estimates are one order of magnitude higher than the mean of 0.002±0.003 Tg/yr estimated for the Baffin Bay and Davis Strait by Manning et al. (2022). Therefore, the Baffin Bay and Davis Strait alone contributed on average 0.3% to the global oceanic $CH_4$ emissions of 9 Tg/yr (Saunois et al., 2020) based on our measurements in 2021.

Atmospheric $CH_4$ mixing ratios during the expedition ranged between 1944.7 ppb off the coast of northern Labrador and 2012.0 ppb in the Cumberland Sound in Nunavut (Fig. 1), with an overall mean (± standard deviation) of 1966.0±7.4 ppb. After filtering measured data, baseline mixing ratios ranged between 1954.2 ppb and 1980.6 ppb (Fig. 6). These concentrations were higher than global monthly mean $CH_4$ mixing ratios in July (1886.4 ppb) and August (1892.6 ppb) of 2021 (Dlugokencky, 2022), but were within range of recent (year 2020) values from surface flask-air measurements from northern stations of the

NOAA Global Greenhouse Gas Reference Network, e.g. Summit, Greenland (July: 1939.2 ppb; August: 1946.7 ppb); Alert, Nunavut (July: 1933.0 ppb; August: 1945.7 ppb); Stórhöfði, Vestmannaeyjar, Iceland (July: 1937.9 ppb; August: 1952.8 ppb); and Ny-Ålesund, Svalbard, Norway (July: 1955.2 ppb; August: 1962.4 ppb) (Dlugokencky et al., 2021). Our measured $CH_4$ values were also consistent with the known latitudinal gradient and recent growth in atmospheric $CH_4$ (Lan et al., 2021). The baseline estimates suggest a local background $CH_4$ fluctuation of roughly 26 ppb in the studied area. A recent study found a

contribution of 42.5±25.2 ppb to total $CH_4$ mixing ratios measured during a cruise in the eastern Arctic Ocean, suggesting that atmospheric $CH_4$ levels over the ocean can be affected by distant wetland $CH_4$ sources (Berchet et al., 2020).





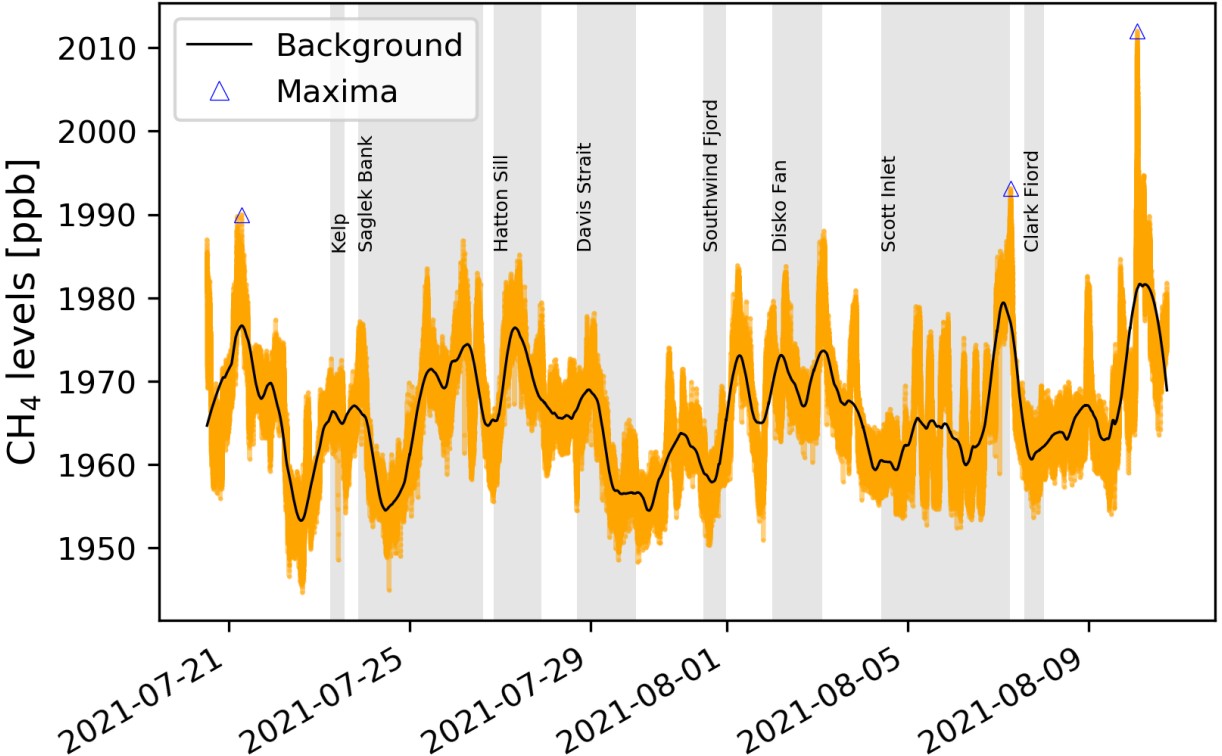

**Fig. 6: Timeseries of atmospheric CH₄ (orange points) and derived background levels (black line) over the entire measurement period. Gray parts show the approximate duration at the stations (Amundsen Science Data Collection, 2021a), where seawater**
**samples were collected. Blue triangles reflect the three maxima of atmospheric CH₄.**

Persistent enhancements of CH₄ mixing ratios above the baseline lasting over roughly 4 hours were detected repeatedly over the length of the expedition (Fig. 6). We investigated potential atmospheric origins of CH₄ maxima at three locations, Cumberland Sound, Scott Inlet, and the Labrador Trough, using ensemble back-trajectories (Fig. 7). At Cumberland Sound, the maximum of 2012.0 ppb coincided with prevailing westerly winds based on our measurements. Therefore, we assumed
that those ensemble trajectories indicating air transport from or across the inland on the western side best reflected the observed meteorological conditions (Fig. 7a). Since no water samples were taken in the Cumberland Sound, where the highest atmospheric CH₄ levels were observed, we could not rule out an ocean-related atmospheric input of CH₄ at this location. Instead, we inferred from a back-trajectory analysis that the elevated CH₄ mixing ratios likely originated from sources onshore such as waterbodies or wetlands. The second highest CH₄ peak of 1993.1 ppb was detected roughly 12 km north-east of the
Scott Inlet seep with dominating easterly winds (Fig. 2, left panel; Fig. 7b). Given the distance of roughly 500 km from Greenland, the origin of this CH₄ enhancement may rather be ocean-based than land-based, which suggests the existence of further seeps along the continental shelf east of Scott Inlet (Gregersen and Bidstrup, 2008; Gautier et al., 2011; Nielsen et al., 2014). Trajectories for the third highest CH₄ levels of 1990.0 ppb measured in the Labrador Trough coupled with west-south-west wind directions suggested onshore sources from northern Labrador (Fig. 7c).





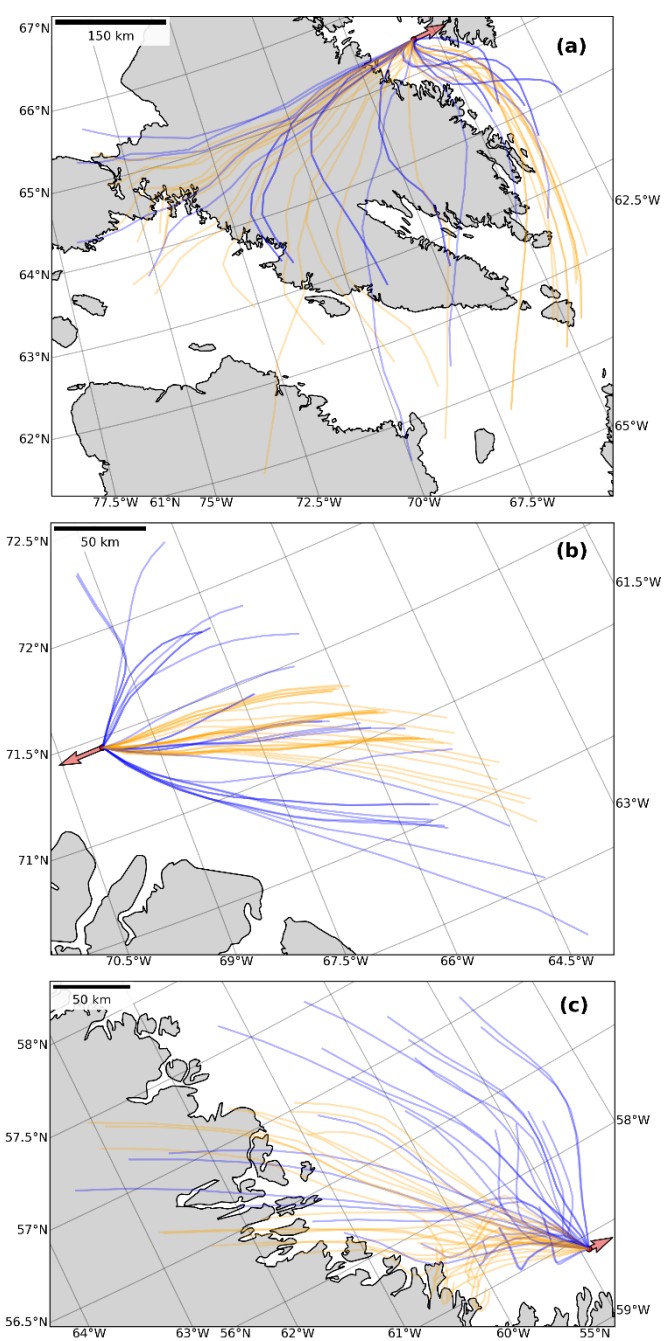

Fig. 7: Back-trajectories of air masses approaching the locations where highest atmospheric CH$_4$ levels were measured in the Cumberland Sound (a), at Scott Inlet (b) and in the Labrador Sea (c). Orange lines represent trajectories using the GFS archive and blue lines show trajectories with the GDAS meteorological model. Red arrows indicate the direction of air movement averaged over three minutes before and after the time of sampling, pointing in the direction the wind is blowing to.





Calculated sea-air fluxes demonstrated that the ocean in the studied area acted as a $CH_4$ source to the atmosphere. However, linear correlations between atmospheric and dissolved $CH_4$ levels based on our dataset were not found which indicates that $CH_4$ released from seeps at the seafloor alone did not directly increase atmospheric $CH_4$ mixing ratios consistent with findings from previous studies (Law et al., 2010; Punshon et al., 2019; Cramm et al., 2021). Furthermore, linear correlations of $CH_4$ mixing ratios with available data were not found, suggesting more complex relationships at sea. Instead, results of a

Generalized Additive Model proposed spatial (latitude, longitude), temporal (hour of day) and meteorological (pressure, dew point temperature) influences on atmospheric $CH_4$ mixing ratios with a good fit (n=173, $R^2$=0.84, 88% explained deviance). In this study, small increments of atmospheric $CH_4$ levels in proximity to seep locations were observed, whereas at other locations where substantial fluxes of $CH_4$ from the sea to the atmosphere were determined locally atmospheric concentrations were not noticeably affected. Therefore, we suggest that atmospheric $CH_4$ levels were influenced by a number of processes

including, but not limited to seafloor seeps, upwind distant land-based sources like wetlands and other waterbodies, weather conditions and ultimately temporal and spatial differences.

## 4 Conclusion

Continuous measurements of atmospheric $CH_4$ levels in remote marine regions of the northern Labrador Sea and Baffin Bay made this study unique. Differences in dissolved $CH_4$ concentrations were mainly affected by ocean currents and seafloor

sources, while atmospheric $CH_4$ levels showed interrelations with environmental conditions, location, and time with small temporal fluctuations. Ocean-based $CH_4$ sources as well as onshore waterbodies and wetlands likely contributed to atmospheric $CH_4$ levels. Further investigation is necessary to confirm potential $CH_4$ sources, for example through analyses of carbon isotopic ratios. We suggested baseline $CH_4$ mixing ratios between 1954.2 ppb and 1980.6 ppb for the studied area which can be used to validate global-scale measurements and modelling. Even though the Arctic Ocean does not contribute significantly

to the global $CH_4$ budget, monitoring and investigation of $CH_4$ levels in and over the sea remains relevant to assess potential impacts of climate change in regions susceptible to permafrost thaw, destabilization of $CH_4$ hydrates and reduced sea ice cover.



**Appendix A**



**Fig. A1: The measurement tower at the bow of the ship with anemometer, temperature sensor, and air inlet mounted on the truss**
**approximately where the arrow is pointing. The GPS was fixed at the lower end of the truss. Photo credit to David Cote (DFO, Canada).**

**Data availability**

Data was made publicly available: Vogt, J., Risk, D., Azetsu-Scott, K., Edinger, E. N. & Sherwood, O. A.: Methane flux estimates from continuous atmospheric measurements and surface-water observations in the northern Labrador Sea and Baffin Bay, https://doi.org/10.5683/SP3/6IUECA, Borealis, 2022.

**Author contribution**

JV, DR and OAS designed and conceptualized the study and JV collected the data. KAS provided the resources for seawater analysis and ENE mentored. JV prepared the manuscript with contributions from all co-authors.

**Competing interest**

The authors declare that they have no conflict of interest.

**Acknowledgements**

We would like to thank the teams from Amundsen Science and the Canadian Coast Guard for their incredible work in preparation of, and during leg 2 of the 2021 expedition. Some of the data presented herein were collected by the Canadian research icebreaker CCGS *Amundsen* and made available by the Amundsen Science program, which is supported through Université Laval by the Canada Foundation for Innovation. We also thank FluxLab members for their support during equipment and data preparation, especially Dr. Evelise Bourlon, Isaac Ketchum and Daniel Wesley. We thank Dr. Carrie-Ellen Gabriel for analyzing the seawater samples, and Dr. Simone Booker, Shaomin Chen and other cruise participants who assisted in the sample collection. Funding was provided by an NSERC ship time grant to Dr. Owen Sherwood and others (RGPST-544990-2020) and a NSERC Discovery Grant awarded to Dr. Owen Sherwood (RGPIN-2018-05590). Seawater analysis was funded through The Aquatic Climate Change Adaptation Services Program (ACCASP) of DFO.

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
