# Peer review of "Methane flux estimates from continuous atmospheric measurements and surface-water observations in the northern Labrador Sea and Baffin Bay"

_EGUsphere, 2022_

## Author Response (AR1)

**Responses to reviewer 1**

*We would like to thank Dr. Charel Wohl for his thorough feedback and his helpful comments on this manuscript. We would like to note that we identified errors in regards to dissolved methane concentration calculations in the manuscript that led to substantial changes in the results of this study upon revision. Therefore, some of the reviewer's comments may not apply to the latest version of the manuscript anymore. In the following, we try to address the comments as best as possible. Our replies to the reviewer's comments are indicated by italic fonts.*

*We would like to note that, upon revision, we made changes (partially beyond the comments of the reviewers) to improve the manuscript. Main changes included:*

- *Correction of dissolved methane concentrations and metadata*
- *Addition of two depth profiles previously excluded because of a misunderstanding*
- *Re-calculation of sea-air fluxes with updated concentrations and slightly different equations*
- *Correction of timestamps in atmospheric dataset and substitution of logged data with analyzer's raw data to overcome data gaps*
- *Exclusion of atmospheric gas measurements accounting for potential contamination by the ship*

**Reviewer's comment on "Methane flux estimates from continuous atmospheric measurements and surface-water observations in the northern Labrador Sea and Baffin Bay" by Judith Vogt**

In this manuscript, the authors describe shipborne ambient air, surface seawater and depth profile measurements of methane in the Labrador Sea and Baffin Bay area. The cruise track focusses in particular on sampling near hydrothermal vents. The authors are investigating the importance of the hydrothermal vent sources . Further, the authors present atmospheric measurements of methane and use air back trajectories to explain peak concentrations in their timeseries. They combine air and water measurements to estimate a sea-to-air flux and investigate latitudinal variations of this flux and the basin source.

Overall, the manuscript is well-written. The value of the manuscript lies in the measurements it presents and the processes it investigates. The manuscript is well referenced making good use especially of very recent publications in the field. I recommend publication after minor corrections. Congratulations on this nice piece of work.

**General comments**

Both these comments are optional and may help improve the readability/intuitive flow of the manuscript.

Maybe the authors could consider reducing the number of display items. The manuscript currently has 7 figures. Maybe figure 2 and figure 7 could be moved to the appendix with a bit more description in the main text instead of these figures.

*Thanks for the suggestion. We moved Figures 2 and 7 to Appendix A.*

Maye the authors could consider presenting the air measurements before the sea-to-air fluxes. It just feels a bit counterintuitive to present fluxes before air mixing ratios as the air concentrations are used to calculate the fluxes.

*We followed the reviewer's suggestion and moved the flux analysis to the end of the Results and Discussion section.*

**Specific comments**

L94: Regarding the linear interpolation of mixing ratios: How long are the data gaps for? What is the longest and shortest data gap here? From the timeseries, I see it is probably very short, but the text in L94 suggests this interpolation was applied to 19 % of the datapoints, which is quite a lot. Maybe consider not doing this linear interpolation for so many points. Instead the authors could set the data to NaN whenever there is a missing value of mixing ratio and report the data in 5 or 10 min (or less) averages. If the authors prefer to proceed with their current data processing, it may be worth highlighting a case study "moment" of how they performed the interpolation as part of the reply to reviewer's comments. Thank you.

*Thank you for your comment. The 19% missing data from the gas analyzer were due to logger-analyzer communication issues. The raw data from the analyzer did not show any gaps. So rather than using the logger data, we fitted the raw analyzer data into the dataset in the revised study to avoid the large number of missing values. We commonly use a ~1 s frequency of our datasets to apply the wind correction.*

L103: . "In addition, we determined there was no significant contamination of air samples by considering $CO_2$ mixing ratios when the air inlet was downwind of the ship's (comparatively elevated) exhaust." That is surprising. Is that because the inlet tower is lower than the ship's stack exhaust? Does that mean that the authors did not do any filtering for ship stack contamination? I.e. the authors also use data when the relative winds were from the rear of the ship or in other words; the air inlet was downwind of the ship's exhaust? Can the authors please provide a figure of mixing ratios of methane and if possible $CO_2$ against the ship's relative wind direction to support their claim? Maybe it is worth filtering the air data and only using methane measurements when the relative winds were from the front of the ship as is common in the field.

*The air inlet was lower than the ship's exhaust. Methane contamination from the exhaust, if present, was not available in sufficient quantities to make a difference for methane measurements. However, we decided to exclude $CH_4$ and $CO_2$ mixing ratios for measured wind directions >80° and <280° relative to the ship's bow, and for $CO_2$ levels >420 ppm to follow the reviewer's suggestion, and also to conform procedures used in other studies. A supporting figure was added in the Appendix.*

L148: It is useful that the authors show exactly what equations they are using from the cited references here. Was there partial sea ice cover during the cruise track? Could the authors please account for this by scaling the flux based on the open water fraction as in https://doi.org/10.1002/2016GL069581 and https://doi.org/10.1002/2017GL073593.

*We added more explanation and used equations to the Appendix. There was marginal sea ice cover during the cruise (also now shown in Figure 1). However, we can confirm that during the Rosette casts when water was sampled no sea ice cover was present. As we are interested in the instantaneous flux of methane (we clarified this in the Methods), we did not account for sea ice cover for the sea-air flux calculations in this study due to its absence.*

Can the authors please provide a few references in which the k for methane has been calculated using the same equation? Maybe it is worth commenting briefly how the parametrisation chosen here compares to other parametrisations of k. This could help explain some of the high fluxes reported in this manuscript.

*We decided to use the same parameterization for k as in the Manning et al. (2022) study (by Ho et al., 2006) instead of the parameterization used in the first version of this manuscript and revised the study accordingly. It should also be mentioned that we found errors in the water analysis, so that methane concentrations were much lower than in the first version of this manuscript. We corrected the values throughout the manuscript.*

L258 It is worth noting here that the fluxes reported in this paper are extremely high for the stations influenced by seeps and North of 65 N. It may be worth emphasising this. The data collected here is clearly biased, because it was collected near known hotspots of methane seeps.

*After correcting errors in the calculations of methane concentrations from the water analysis, fluxes were much lower than in the previous version of this manuscript and match better with previous studies.*

L268 Diligently thinking ahead, the authors scale their measurements up to the whole area and exclude "samples with high concentrations". Can the authors please clarify how they have carried out this filtering?

I highly recommend redoing this calculation and accounting for sea ice cover and assuming sea ice acts as a barrier to air-sea exchange. Maybe the author's estimate would nudge closer to the one presented in Manning et al. (2022).

*We decided to remove the calculation of a basin-wide flux from this study and focus on instantaneous fluxes.*

Furthermore, I note that Manning et al. (2022) and the manuscript under review here use different air sea gas transfer parametrisations for k. This could be another reason why the estimate in this manuscript is much higher than the one from Manning et al. (2022). I am no expert on air sea gas transfer parametrisations for methane and cannot judge which one may be preferable. In the flux methods section, the authors could further defend their choice of air sea gas transfer parametrisation.

*Generally, the use and choice of the k parameterization remains debatable and its use in different studies varies. All parameterizations come with uncertainties. We decided to use the same parameterization of k for flux calculations as in Manning et al. (2022) to generally make comparison easier.*

Overall, I don't think it is a very good idea to extrapolate on the Baffin Bay and Labrador sea source from the measurements in this manuscript. The measurements in this manuscript are biased to sample in high concentration areas and extrapolation thereof is clearly going to lead to very high contributions/emission fluxes. I suggest the authors delete this section or focus on estimating the contribution of methane seeps to the Arctic methane emission budget. The dataset would be better geared for that. Alternatively, the authors can also keep this section with the modifications suggested above, but I highly recommend adding an explanation that the

sampling here was carried out near methane hotspots and is thus biased towards high emission fluxes.

*We decided to remove this section from the manuscript.*

L291 In this section, the authors use back trajectories to explain three peak concentrations observed during the cruise track. Back trajectories are often used in atmospheric measurement papers, but they have also limits and uncertainty associated with them. The authors clearly also show that they get slightly different results based on what model or archive they use. In this paragraph, there are only three back trajectory runs shown and the conclusions are not very convincing. The authors could consider removing this section. To further strengthen their argument, the authors may want to analyse where their lowest methane concentrations come from or do some more complex back trajectory analysis including multiple runs.

*From an atmospheric science point of view, we think the back-trajectory analysis gives a valuable insight into the possible movement of air parcels in this study. The reviewer is right though, that this analysis comes with uncertainties. We were mainly interested to see if air rather moved from land or the open ocean towards the point of measurement. The air may come from along the lines of the trajectories, but we acknowledge that discrepancies to the model are possible. We generally amended the statements in this section, and the respective figure was moved to the Appendix.*

L298 "Instead, we inferred from a back-trajectory analysis that the elevated CH4 mixing ratios likely [use "maybe" here] originated from sources onshore such as waterbodies or wetlands." This sentence would benefit from a reference or an additional explanation that waterbodies and wetlands can be sources to the atmosphere. Following on from the previous comment as well, this statement is not very well supported by showing air back trajectories alone. Just because the air travelled over areas of potential sources cannot explain that the high mixing ratios. Was the source strong enough to change air mole fractions? Did the air mass spend enough time near the source? The latter two questions are just to stimulate thought on this topic.

*We added a reference to support our statement and softened it as suggested.*

**Detailed comments**

Title: Maybe Change the title to emphasise the depth profiles. A large part of the results and discussion is dedicated to this though it is not mentioned in the title. There is also no mention of the cold seeps in the title. Including this may increase readership.

*We changed the title.*

L 11: I suggest to remove permafrost from the abstract and maybe replace it by hydrothermal vents? The manuscript focusses on hydrothermal vents, not on permafrost.

*We removed "permafrost" from the abstract.*

L 15: Here and throughout; Remove "real-time". I don't think this is the right word. The authors may be thinking of "high-resolution"? Maybe also remove "along with ambient air temperature and wind parameters" as these are routine measurements.

*We followed the reviewer's suggestions.*

L 17: "up to 71°N", give full lat rage of cruise

*We revised this and state the full latitudinal range.*

L 18: Here and throughout: Why highlight "selected stations"?

*We revised this and use "various stations" instead.*

L 18: Note that ppb is not an SI unit and can be misleading. Maybe consider replacing it by "ppbv" or adding a sentence to clarify that the authors mean "nmol mol-1" with ppb(v) in this manuscript.

*We replaced "ppb" by "ppbv" and "ppm" by "ppmv" throughout the manuscript.*

L19: Maybe worth adding a line somewhere to clarify the authors mean "nmol dm-3", when talking about "nM" as the latter is also not strictly speaking an SI unit.

*We replaced "nM" with "nmol/L" in the abstract and made a note in the Methods section.*

L21: Where the calculated fluxes always from sea to air? If so, maybe worth highlighting this in the abstract with "consistently supersaturated"

*We added "at all stations" to emphasize that all sea-air fluxes were positive, i.e. from sea to atmosphere.*

L23: "Highest atmospheric CH4 mixing ratios were detected in the Cumberland Sound in Nunavut, suggesting onshore sources from nearby waterbodies and wetlands, whereas ocean-based contributions at this location could not be ruled out." This is a bit of a weak conclusion as the authors highlight in the same sentence as well. Maybe this is not essential to highlight in the abstract? It would be fine to leave it if the authors prefer so. Replace "whereas" by "however".

*We removed the sentence.*

L30: "radiative activity" – Do the authors mean "radiative-forcing"?

*Yes. We revised that. Thank you.*

L31: "determine" – Do the authors mean "immediately detect"?

*We followed the reviewer's suggestion.*

L33: "The Arctic Ocean contains large amounts of CH4 in sediments along the continental margins." Reference required.

*We added references.*

L40: "dissolution" hints somewhat at solubility. Suggest changing to "distribution in surface waters and in the water column".

*We followed the reviewer's suggestion.*

L 42: Typo? Two times CH4 in same sentence.

*We revised the sentence accordingly.*

L50: "mobile" – unusual choice of words. Suggest remove.

*We removed "mobile".*

Figure 1: In the methods sections, the authors are showing their methane air mole fractions, which are results. Consider maybe colouring the cruise track by sampling date as the cruise track is not straight forward. Beyond that, overall a great figure. Can the authors maybe indicate sea ice in this map as faint shading or would this interfere with the bathymetry colouring?

*We moved the figure into the Results section. We want to avoid repetition of the figure (showing cruise track in the methods and methane mixing ratios in the results). We retrieved satellite-based sea ice data (EUMETSAT), and added the small area of sea ice cover >10% in the map.*

L 61: Before mentioning the seafloor gas seepage points, maybe worth highlighting that they are indicated in the map.

*We added a reference to Figure 1.*

L65: Maybe consider shortening the description of currents and reduce it to what is necessary for the understanding of the manuscript? Or move this description to later on in the manuscript where the authors are talking about the latitudinal flux gradient.

*We shortened the description.*

L 93: "over a distance of 5100 km between July 20, 2021, and August 10, 2021," -unnecessary repetition, maybe delete.

*We removed "over a distance of 5100 km" but kept the dates since they slightly differ from the dates of the cruise allowing time at the beginning and the end to set up and take down instruments.*

L99: Please review the references of "(Amundsen Science Data Collection, 2021c)". It seems C is before A and B in the main text.

*The references were reviewed.*

L101: "benchmarking" is that the same as blanking the instrument or determining instrument background? If so, "benchmarking" is an odd choice of words.

*For atmospheric greenhouse gas measurements, benchmarking is performed to verify drift of the analyzer and it is a commonly used term. During the process of benchmarking, we exposed the analyzer to the same dry air on a daily basis to determine a potential drift of the analyzer.*

L121: Just as a suggestion; Maybe the authors could consider naming the CTD stations numerically or alphabetically as well as the "station name" given already. Maybe this would make it more accessible for a reader less familiar with the station names.

*We acknowledge the suggestion, but want to avoid confusion of missing numbers/stations for example for the depth profiles that were not taken at all stations listed here.*

Figure 3: This figure needs a legend for the blue and black dashed line. "A legend should clarify all symbols used and should appear in the figure itself, rather than verbal explanations in the captions (e.g. "dashed line" or "open green circles")." See https://www.biogeosciences.net/submission.html#figurestables. This comment applies to all figures in this manuscript (also the map etc.). Maybe the marker size could also be increased to make it clearer at what depths measurements were taken.

*We added a legend, increased the markersize and revised the figure.*

Is there oxygen data available from the mixed layer which could be used to explain interannual variability in surface water concentrations? Some of the interannual variability in surface water concentrations could be due to methane oxidation rates. The authors present the importance of methane oxidation in controlling concentrations later in the text.

*After the correction of the concentrations, we did not observe strong interannual variability anymore. However, we retrieved oxygen data from CTD measurements for 2021, but that data did not support our hypotheses that higher oxygen levels could be found where methane concentrations were low. Instead, we found a positive relationship between mean oxygen and methane levels within the mixed layer for most stations and included this finding in the Results.*

Maybe some indication of the mixed layer depth would be useful in these figures to illustrate the role of mixed layer bacterial activity in reducing concentrations near the surface.

*We added the (generally very shallow) mixed layer depth determined from CTD data in the figure and compared the relationship of mean methane and oxygen levels among stations, which showed a positive relationship as mentioned above.*

L159: Please add a space before "%" throughout. See "Spaces must be included between number and unit (e.g. 1 %, 1 m)." https://www.biogeosciences.net/submission.html

*We revised the manuscript accordingly.*

L209: Overall this discussion is well written and suggests large methane concentrations near seeps and large inter-annual variability of these high concentrations.

*These results changed during the revision.*

Figure 4: The presence of the seeps in the dataset makes it very difficult to extract much useful information from this figure on what is "generally" happening in the study area. In the figure, no obvious trends are visible. Maybe the authors could "hide" or "highlight" points potentially affected by the seeps. This allows them to comment on what is generally happening in the area or what water masses are most affected by the seeps. Apart from that, the discussion of the figure is very clear.

*After revising the error in the water sample analysis, the fluxes match those from previous studies better. In addition, we highlighted the points in proximity (within 50 km) of the Scott Inlet seep, which comprise the highest methane concentrations over the years.*

L242: The figure reference to Fig. 5 disrupts the flow a bit here. Consider removing it and moving the first mention to later in the text. Maybe even just the next sentence where the difference 65 N and S is discussed.

*We revised accordingly and moved the reference to the next sentence.*

L245 Maybe it would be worth indicating the Labrador current in Figure 1.

*We added an arrow for the Labrador Current in the figure.*

L246 It is hinted here a little bit, but I was wondering if the authors had considered to look at sea ice cover as a controlling factor of methane concentrations. Sea ice cover controls factors such as bacterial activity (10.1525/elementa.2020.00113), mixed layer depth (10.1525/elementa.372) and air-sea flux (https://doi.org/10.1002/2016GL069581 and https://doi.org/10.1002/2017GL073593). These should also influence dissolved methane concentrations. See 10.1038/srep16179

*We did not directly collect water samples in areas with sea ice cover. There was some sea ice cover at other locations during the cruise, but water sampling was not performed there and we made clear that we determined instantaneous sea-air fluxes in this study.*

Figure 5: Similar to figure 3, a figure legend is missing.

*We added a legend.*

L260 "featuring large uncertainties" Can the authors please elaborate on this? What is the source of these uncertainties?

*Sorry for the confusion. We referred to the large uncertainties of the mean, ie. the high standard deviations of the sea-air fluxes stated in the text. However, these results changed after the revision of methane concentrations and fluxes.*

L269 "in which case the ocean may act as a small CH4 source or sink to the atmosphere" I thought the fluxes measured during this cruise were always out of the ocean? Do the authors suggest this because the standard deviation here is larger than the mean? A larger standard deviation than the mean could also be because the data is skewed and the mean is influenced by outliers of high concentration.

*Correct, we suggested that since the standard deviation was higher than the mean. As already mentioned, the values changed after the revision of methane concentrations and fluxes lie within a closer range.*

L271 Delete "therefore" and replace by: Using our estimate of the contribution of Baffin Bay and Davis Strait to the Arctic Ocean methane source, …

*We removed this section from the manuscript.*

L277 "After filtering measured data," See an earlier comment. It is not very clear how this filtering was done.

*Here we applied a Savitzky-Golay filter after Savitzky and Golay (1964) of second polynomial order as mentioned in the Methods. We clarified that we used this filter here.*

L278 "of 2021" – of the sampling year 2021 and L279 "but were within range of recent (year 2020) values from surface flask-air measurements" change to "but were within range of values from surface flask-air measurements from the year 2020"

*We followed the reviewer's suggestion.*

L280 The authors are comparing their methane measurements from Baffin Bay area to many other stations in the Arctic. I suggest comparing the author's measurements only to the closest station (Alert?). Doing so, illustrates that the concentrations measured in this deployment are higher than that station's mean. This is likely because the cruise track focussed on sampling air near methane seeps and thus emission hotspots.

*We removed the Iceland and Norway flask samples but consider the Nunavut and Greenland stations reasonable to mention, also given that they are about the same distance to the cruise track. We implemented the reviewer's interpretation to clarify the use of mentioning the flask samples.*

L283 "growth" replace by increase

*We followed the reviewer's suggestion.*

L310 Please split this sentence in 3 to enhance readability. Further, the lack of a correlation between water concentration and air mole fraction is not an indication "that CH4 released from seeps at the seafloor alone did not directly increase atmospheric CH4 mixing ratios". Surely the situation is similar to dimethyl sulfide where highest air mole fractions are not observed above areas of highest seawater concentrations, because air parcels move much faster than water parcels and other complexities (differences in lifetime in air and water, air-water interface acting as a barrier and thus the two are decoupled, different sources and sinks in air and water as the authors discuss in the manuscript etc.). E.g. https://bg.copernicus.org/articles/19/701/2022/bg-19-701-2022.pdf, page 710

*We split the sentence, revised the text with updated fluxes and incorporated the reviewer's suggestion and reference.*

L313 "Furthermore, linear correlations of CH4 mixing ratios with available data were not found, suggesting more complex relationships at sea." Change to: "Furthermore, simple linear correlations of CH4 mixing ratios with available atmospheric auxiliary data were not found, suggesting more complex relationships." Maybe briefly mention what atmospheric auxiliary data was tested for here.

*We followed the reviewer's suggestion and elaborated on which data was used for the linear regression.*

L315: Why is the n number so low here? Was there data excluded from the fit?

*The number n is so low because atmospheric pressure and dew point temperature data from the ship's own measurement system had large gaps. For the GAM, we used hourly averages, which already reduced the number of datapoints to 510. Therefore, only data where atmospheric pressure, dew point temperature and methane mixing ratios were present could be used for this model.*

L317: "In this study, small increments of atmospheric CH4 levels in proximity to seep locations were observed, whereas at other locations where substantial fluxes of CH4 from the sea to the atmosphere were determined locally atmospheric concentrations were not noticeably affected" – replace by "In this study, small increases of atmospheric CH4 levels in proximity to seep locations were observed. At other locations where substantial fluxes of CH4 from the sea to the atmosphere were determined local atmospheric concentrations were not noticeably affected" Did the sea-to-air flux correlate with methane air mixing ratios? "small increments of atmospheric CH4 levels in proximity to seep locations" – this is not very clear from figure 3. The air mixing ratios do not show a clear trend because air parcels travel fast and the sampling location (upwind or downwind from seep) also plays a role as well as the air sea flux. The authors should be more clear what data they are referring to which supports this claim.

*We removed this paragraph after updating the sea-air fluxes. Sea-air fluxes and atmospheric methane mixing ratios did not correlate.*

L323 "Continuous measurements of atmospheric CH4 levels in remote marine regions of the northern Labrador Sea and Baffin Bay made this study unique." Suggest changing this sentence. Are the seawater measurements, depth profiles and flux calculations of value as well? Especially since they represent measurements near methane seep sites. Additionally, the year-on-year comparisons in Fig. 3 are probably of value as well. Personally I am quite impressed with the large methane fluxes from the seeps. The fact that these fluxes are so large, probably warrants monitoring indeed, as the authors suggest. The authors could review the conclusion from their manuscript to better highlight the importance of their manuscript.

*We adjusted the conclusion to better highlight the importance of the manuscript.*

Appendix A. Great picture.

*Thank you!*

**Responses to reviewer 2**

*We would like to thank the reviewer for their thorough and constructive feedback. We apologize for the flaws in the first version. During the revision, we encountered errors in the calculation of dissolved methane concentrations. These errors were fixed and the results changed in the revised manuscript. We tried to answer and address the reviewer's comments and suggestions as best as possible. Our answers are indicated in italic below.*

*We would like to note that, upon revision, we made changes (partially beyond the comments of the reviewers) to improve the manuscript. Main changes included:*

- *Correction of dissolved methane concentrations and metadata*
- *Addition of two depth profiles previously excluded because of a misunderstanding*
- *Re-calculation of sea-air fluxes with updated concentrations and slightly different equations*
- *Correction of timestamps in atmospheric dataset and substitution of logged data with analyzer's raw data to overcome data gaps*
- *Exclusion of atmospheric gas measurements accounting for potential contamination by the ship*

**Review of Vogt et al. (2022)**

**General comments:**

This paper presents atmospheric methane and dissolved methane measurements from a research cruise in Baffin Bay and Davis Strait in summer 2021. The stations where water samples were collected contain several stations with known benthic methane sources (e.g. Scott Inlet) as well as some stations with methane levels more typical of the broader region.

I first want to acknowledge the efforts of the PhD student who led the paper. They have clearly worked hard to prepare this manuscript.

However, the paper contains many major errors that need to be addressed before the paper is published. It is apparent to me that the student has not received sufficient mentorship and guidance on interpretation of the data from scientists with expertise in dissolved gas measurements, sea-air flux measurements, and Arctic Oceanography. The student appears to be working in a research group that focuses on gas emissions from industrial sources. The senior scientist coauthors of the paper who do have oceanographic expertise have clearly not read the paper thoroughly. There are fundamental errors such as the wrong units being used for ocean salinity (figure 4), errors in the gas solubility calculations, flux calculations, and error estimates that will substantially alter the results once corrected.

For this article to be publishable, the coauthors with appropriate expertise need to take an active role in helping the student to correct these errors, and the student and their supervisor may need to establish a collaborative relationship with researcher(s) with expertise in oceanographic sea-air flux calculations who would become coauthor(s) on the manuscript to correct these major errors. A peer reviewer should not be identifying such fundamental issues.

**Specific comments:**

**Errors in calculated methane saturation states (line 158)**

Line 158: "Seawater samples showed wide ranges of dissolved CH4 concentrations at the different sample locations and water depths from undersaturated (53%, 0.2 nM) to highly oversaturated (6858%, 272.4 nM, Fig. 3)."

The solubility of methane in Arctic seawater is ~4 nM. A 53% saturation is equivalent to roughly 2 nM , not 0.2 nM,. The highly oversaturated value in this sentence contains a similar error. I am unsure if the saturation states or concentrations are incorrect, and I don't have time to figure out how many similar errors are present in the manuscript and if these errors extend to their flux calculations. The concentrations and saturation states shown in Figure 3 appear to be approximately correct (100% saturation corresponding to 3-4 nM.)

This is a major error that needs to be carefully reviewed throughout the manuscript and underlying calculations.

*The reviewer is correct, the numbers were incorrect. Unfortunately, we found mistakes in the calculation of CH4 concentrations in the water sample analysis. We apologize for this flaw. In brief, most CH4 concentrations were wrong and generally numbers were much lower after revision, so that the storyline of the study changed. We revised the text and figures accordingly.*

**Errors in the gas transfer velocity equation and sea-air flux calculations (line 154)**

The flux equation on line 154 requires the wind speed to be U10, the wind speed at 10 m height above sea level. It appears that they have directly used wind speeds from the anemometer which was reported to be 8.1 m above deck (line 82) and likely significantly more than 10 m above sea level based on Fig A1. They need to adjust the wind to 10 m height using one of many published methods.

*It is correct that the anemometer was mounted at a location higher than 10 m above sea level, approximately at 14 m above sea level. It should be noted that due to vertical motion of the ship but also depending on the ship's load, the height of the anemometer above sea level was technically not constant. To conform with standard calculations, we corrected the wind speed to a 10 m above sea level via Power Law.*

They also need to correct the fluxes for sea ice cover as they claim on line 64 "The studied region lies within the seasonal sea ice zone and the ocean was partially covered with sea ice in the northernmost regions."

*We can confirm that during the water sampling with the CTD-Rosette at the stations in this study no sea ice was present. Sea ice was present at other locations during the cruise, though. We clarified that in the text. We were interested in instantaneous sea-air fluxes and therefore disregarded the fact that there was partial sea ice cover at times when no water was sampled during the cruise.*

It is surprising to me that they "determined there was no significant contamination of air samples by considering CO2 mixing ratios when the air inlet was downwind of the ship's (comparatively elevated) exhaust" and would like to see this data for both CO2 and CH4. I also recommend they contact other researchers who have made atmospheric CO2 and other gas measurements from the flux tower on the CCGS Amundsen in prior years and inquire if they observed contamination from the ship's exhaust and how they detected it.

*Methane contamination from the exhaust, if present, was not sufficient to make a difference for methane measurements. However, we decided to exclude CH4 and CO2 mixing ratios for measured wind directions >80° and <280° relative to the ship's bow, and for CO2 levels >420 ppm to follow the reviewer's suggestion, and also to conform to procedures used in other studies. A supporting figure was added in the Appendix.*

They claim the Schmidt numbers were taken from Wanninkhof (2014). Did they correct the Schmidt numbers for salinity? The values should be interpolated from the the freshwater (S=0) and seawater (S=35) values, based on the salinity of the sample (often significantly less than S=35 in Arctic surface waters). Also, note that Wanninkhof recommends authors cite and consult the original sources for the Schmidt numbers.

*The Schmidt numbers were not corrected for salinity in the first version of this study. After revision of the methane concentrations, fluxes were re-calculated using code examples from Manning & Nicholson (2022), which consider a correction for salinity according to Jähne et al. (1987).*

Furthermore, I believe it is very misleading to calculate an annual flux over the whole study region based on their data, which is based on a 3-minute shipboard measurement of wind speed and has significantly higher concentrations than prior studies at similar stations, and is from a cruise where more than half of the surface samples were specifically targeting known methane seeps or otherwise had elevated levels.

As discussed in Wanninkhof et al. (2009) and other studies, due to the nonlinear dependence of gas flux on wind speed, flux estimates based on instantaneous winds will be systematically biased compared to longer-term winds. Furthermore they calculate a flux "Considering all measurements from 2021 and an area of 1,123,000 km² for the Baffin Bay and Davis Strait" but this seems inappropriate when the majority of stations were from known methane seeps and 5 out of 13 measurements were collected within 5 km of a known methane seep (if I am understanding their methods correctly); these sites are extreme outliers (as shown in Manning et al., 2022 and Punshon et al. 2019 among others).  They also calculate a flux "if samples with high concentrations were excluded" but don't explain how 'high concentrations' were defined, and this likely brings the number of stations down to a very small number (perhaps 5 stations) which is not appropriate to extrapolate over a larger region.

As they do not account for seasonal and annual variability in the gas transfer velocity (due to ice and wind speeds), their annual fluxes based on 3 minutes of wind speed data are very unlikely to be accurate. Their surface concentrations are much higher than reported in previous studies at the same/similar stations (e.g. Cramm et al., 2021, Manning et al., 2022, Punshon et al., 2019, etc.) which I believe makes it even less appropriate to calculate a flux based on their data only.

*We understand this concern and excluded the calculation of annual fluxes from the analysis in the manuscript.*

**Uncertainty estimates are 1-2 orders of magnitude too small (line 130)**

The authors need to clarify whether any duplicate samples were collected. It appears the answer is no. They claim "Uncertainty in dissolved CH4 was ±0.08% (Punshon et al., 2014, 2019)." World-leading labs typically report combined uncertainty (incorporating accuracy and precision of replicate measurements) of one to several percent for dissolved methane analysis

in seawater. I took the time to review Punshon's 2019 paper which claims "Analytical precision, determined by repeated analysis of air equilibrated seawater, was ± 0.80% for methane." Analytical precision of air equilibrated water samples prepared in the lab is not the same as the combined/propagated uncertainty of field samples.

It is misleading to claim in the abstract (line 20) that "Dissolved CH4 concentrations in the near-surface water peaked at 56.58±0.05 nM" – this appears to be based on erroneously applying an accuracy of 0.08% (0.05 nM) to a single bottle measurement. If there were no repeat measurements during this cruise, then in my opinion they should be upfront about this and not apply an uncertainty that does not account for all sources of error.

*We did not collect repeat measurements during the cruise (with one exception) and corrected the uncertainty statement in the Methods and refrain from using uncertainties for the individual methane concentrations in the revised version.*

**Technical comments and recommendations:**

Figure 4: Salinity is incorrectly labeled as "permil"

*We corrected the units to psu.*

Dataset– I downloaded the data and noted they have put both the dissolved data and atmospheric data into the same file. There are 1826786 rows in the file and only 13 of them contain dissolved gas and flux data. I recommend they make a separate file that contains the dissolved gas data only. They also need to archive the exact depth/pressure, salinity, temperature data corresponding to each dissolved gas sample to ensure reusability of the data and reproduce figures in the manuscript. Currently they have only archived the concentration and approximate/target depth, e.g. 10 m.

*We split the dataset into atmospheric and dissolved methane measurements, and added relevant CTD data.*

The final sentence of the manuscript is misleading: they state "the Arctic Ocean does not contribute significantly to the global CH4 budget." This manuscript does not prove or disprove this statement; the authors only collected instantaneous data in one small region of the Arctic and subarctic. In the intro, line 42, they report the very high CH4 fluxes reported from the East Siberian Arctic Shelf as a motivation for this study.

*We clarified that this statement was not based on findings from this study.*

Line 87: Where were the calibration gases obtained? What was the accuracy of the cylinder actually used in the field?

*For calibration, we used Ameriflux standard gas. The cylinder used in the field was filled with CO2 mixed with ambient air, so that methane concentrations were not determined, but amounted to ~5 ppm with unknown uncertainty. Using this cylinder in the field was not planned, but due to some logistical issues, it was the only option for benchmarking. However, the analyzer was calibrated before using it in the field, and the benchmarking serves the purpose of detecting analyzer drift, which can be determined from any gas containing methane as long as its concentration does not change over time. The uncertainty of the gas is not crucial in this case.*

They need to clarify throughout manuscript and in published dataset if the atmospheric mixing ratios are based on dry mole fraction or include the water vapor pressure.

*We used dry mixing ratios and clarified that in the Methods section.*

Latitudes and longitudes require a degree symbol. Longitudes reported as W do not need a negative sign.

*We revised that.*

Fig 3: Are the profiles from 2012 and 2016 the CH4 saturation or CH4 concentration? I suggest making an updated legend that would show each line type, the year, and whether it is concentration or saturation.

*We added a legend for clarification and included more nearby stations to account for all stations within 50 km of those visited in 2021.*

Fig 5: Add a legend to accompany the caption.

*We added a legend.*

I do not have expertise in atmospheric modeling, and I am not qualified to comment on whether there are errors in these aspects of the manuscript.

**Final notes**

I want to again commend the PhD student on their strong (and likely quite independent) effort. I again emphasize that I believe the issues with this manuscript arise from a lack of engagement from the more senior coauthors and/or lack of collaboration with researchers with necessary expertise. Early career researchers require this mentorship and collaboration to develop advanced level scientific knowledge.

**References**

Cramm, M. A., Neves, B. de M., Manning, C. C. M., Oldenburg, T. B. P., Archambault, P., Chakraborty, A., et al. (2021). Characterization of marine microbial communities around an Arctic seabed hydrocarbon seep at Scott Inlet, Baffin Bay. Science of The Total Environment, 762, 143961. https://doi.org/10.1016/j.scitotenv.2020.143961

Manning, C. C. M., Zheng, Z., Fenwick, L., McCulloch, R. D., Damm, E., Izett, R. W., et al. (2022). Interannual Variability in Methane and Nitrous Oxide Concentrations and Sea-Air Fluxes Across the North American Arctic Ocean (2015–2019). Global Biogeochemical Cycles, 36(4), e2021GB007185. https://doi.org/10.1029/2021GB007185

Punshon, S., Azetsu-Scott, K., Sherwood, O., & Edinger, E. N. (2019). Bottom water methane sources along the high latitude eastern Canadian continental shelf and their effects on the marine carbonate system. Marine Chemistry, 212, 83–95. https://doi.org/10.1016/j.marchem.2019.04.004

Punshon, S., Azetsu-Scott, K., & Lee, C. M. (2014). On the distribution of dissolved methane in Davis Strait, North Atlantic Ocean, 161(C), 20–25. https://doi.org/10.1016/j.marchem.2014.02.004

Wanninkhof, R., Asher, W. E., Ho, D. T., Sweeney, C., & McGillis, W. R. (2009). Advances in Quantifying Air-Sea Gas Exchange and Environmental Forcing. Annual Review of Marine Science, 1(1), 213–244. https://doi.org/10.1146/annurev.marine.010908.163742

Wanninkhof, R. (2014). Relationship between wind speed and gas exchange over the ocean revisited. Limnology and Oceanography: Methods, 12(6), 351–362. https://doi.org/10.4319/lom.2014.12.351

---

## Author Response (AR2)

**Responses to the reviewer**

*We would like to thank the reviewer for their thorough feedback. We answer the reviewer's comments below, indicated by italic text.*

I wish to acknowledge the work done by the authors to revise the manuscript. I am glad that my review enabled you to identify major errors in the dataset that could be corrected before publication. I believe this is an example of the peer review process working. Given the manuscript was completely rewritten it took time for me to carefully review the new work. I appreciate your patience.

*Thank you for re-revising the manuscript.*

My biggest remaining with the manuscript is that they have not adequately explained how the atmospheric CH4 analyzer was calibrated and quality was ensured. They state it was "benchmarked daily" but do not explain whether the benchmark data was used for any drift correction of the data . The composition of the gas is also confusing.

The author's response (page 14) states "The cylinder used in the field was filled with CO2 mixed with ambient air, so that methane concentrations were not determined, but amounted to ~5 ppm with unknown uncertainty. Using this cylinder in the field was not planned, but due to some logistical issues, it was the only option for benchmarking."

The revised manuscript, line 86 says the analyzer was "benchmarked daily (except for the first two days due to logistical issues) with a certified standard gas mixture (from Praxair) of 450 ppmv CO2 balanced with air containing 5000 ppbv CH4, which was well within the analyzer's measurement range".

The first statement implies the final concentration was 5 ppmv and the second implies that the original concentration before dilution was 5 ppmv. Considering that the concentration of CH4 in ambient air is 2 ppmv, I don't understand how a standard gas diluted with ambient air could end up with a final concentration of 5 ppmv CH4 (2.5x the concentration of ambient air). What was the concentration of CO2 in the final mixture (based on dilution this would tell us what was the original combination in the ambient air)?

I would suggest rewording it as "benchmarked daily (except for the first two days [list dates here] due to logistical issues) using a gas that was a mixture of a 450 ppm CO2 standard from Praxair and air containing roughly ZZZ ppbv CH4. The concentrations of CO2 and CH4 measured in the benchmark gas over the whole cruise were X+/-x and Y+/-y (mean +/- standard deviation), respectively. Add text here quantifying the drift in CH4 and CO2 observed during the cruise on the benchmark gas and explaining whether the benchmark gas data was used to perform any post-corrections to the atmospheric data or simply as a check to see whether drift was detectable. Add a statement that a post calibration with a certified reference gas was not done and not used to post-correct for drift (I think this is correct but I am not sure). State the date that the sensor was calibrated (currently it is stated as July 2021 and the cruise stated July 12).

*Thank you for your suggestion. The gas cylinder's certificate indicated a concentration of 450 ppmv for CO2. It was not certified for CH4. Only based on our measurements, we determined a concentration of*

*about 5 ppmv. We clarified this in the text and followed the reviewer's suggestions by adding more details about the benchmarking procedure. We would like to note that the benchmarking was most suited to determine whether the analyzer and the setup was functioning properly, to some extent also to determine how precise measurements were during each benchmarking, and it may also be used to determine drift. However, benchmarking during field campaigns often shows a larger drift in the measurements than what we yield from in-lab tests. This may be attributed to different ambient conditions, vibration, power supply, etc. that could slightly affect measurements in the field, but these conditions are more controlled and difficult to reproduce in-lab. A post-expedition in-lab benchmarking was conducted, on which we based our decision not to post-correct our data. We also included this information in the text.*

Other comments:
In my opinion, the current title is misleading: "Sea-air methane flux estimates derived from continuous atmospheric measurements and marine depth profiles in cold seep regions"

The previous title was somewhat better: "Methane flux estimates from continuous atmospheric measurements and surface-water observations in the northern Labrador Sea and Baffin Bay"

I believe the current title is misleading because
A. The depth profiles are not used to calculate fluxes – only the surface measurements
B. The continuous observations are not used to calculate fluxes: you take a 10 min average coinciding with the surface water measurements, as indicated on line 155. If I only read the title I would assume that you had used the eddy covariance technique to calculate fluxes.

I would recommend a title such as "Continuous atmospheric methane measurements and marine depth profiles in the northern Labrador Sea and Baffin Bay" OR
"Sea-air flux estimates derived from marine surface measurements and instantaneous atmospheric measurements in the northern Labrador Sea and Baffin Bay"

*Thank you for the suggestion. We changed the title to "Sea-air methane flux estimates derived from marine surface observations and instantaneous atmospheric measurements in the northern Labrador Sea and Baffin Bay".*

line 76: "Meteorological Tower" and "Model" should not be capitalized

*We revised that.*

Line 132: "The analytical precision was estimated from repeat measurements of standard gases and amounted to 0.5–0.8 % or better for dissolved CH4 similar to previous studies (Punshon et al., 2014, 2019)." It is not appropriate to report analytical precision based on standard gases. You need to report precision based on samples, which have a completely different workflow and likely a lower precision.

*We only took one sample per depth and location and only analyzed each sample once. This means we did not report the precision based on samples. What we reported here was analytical uncertainty of the instruments. We reworded the phrase to clarify this and added a statement that replicate samples were*

*not taken.*

Line 134: change "and density" to "and potential density"

*We revised that.*

Line 144: "we also used seawater density and oxygen data" – but no oxygen is presented in the paper, nor in the data.

*We do not show any figures including the oxygen data, but determined oxygen levels in the mixed layer and report a statistical relationship in the Discussion. Therefore, this data is mentioned in the Methods, but we don't think that it requires inclusion in the dataset, also because we don't present any other mixed layer data there. We added a few words to the text clarifying the use of oxygen data in this study and that the data is not shown further.*

Line 270: "The mixing ratios measured in this study are higher than those determined from flask samples likely due to the influence of CH4 seeps in our study area."
I think it is misleading to claim this is the "likely" cause without any data demonstrating differences in methane fluxes between the regions where the flasks were collected and where your study occurred. The observed near-surface saturations of roughly 110 to 150% are similar to other Arctic seawater measurements. You state earlier in the article that nearly all of the seep-derived methane was oxidized before making it to the air-sea interface. What if the difference is just a calibration offset?

*We revised the statement in the text. Flask samples usually undergo strict protocols to assure the correctness of gas mixing ratios. Our gas analyzer was calibrated prior to deployment and the "goodness" of this calibration was confirmed by benchmarking afterwards in-lab. Therefore, we are confident to state that flask samples showed lower mixing ratios compared to our measurements.*

Data: I mentioned in my previous that the exact depth needs to be archived and this has not been fixed – it is still listed as the approximate depth e.g. 100 m, 800 m, etc.

*We revised the depth and updated associated figures and the dataset.*

Rows 76-78: three samples from three different locations are reported with exactly the same temperature and salinity (SE-1K, NE-1K, NE-5K). Is this a typo? I assume that you can obtain T and S data from the underway thermosalinograph if a rosette was not deployed.

*Thanks for catching this. We used available data from the Rosette casts to fill missing data, revised the dataset and updated it accordingly. For your information, we carefully double-checked all salinity, pressure, temperature and depth data and found a few more typos which were revised, not noticeably affecting the results. We also revised the manuscript accordingly.*